# Estimating Topic Modeling Performance with Sharma–Mittal Entropy

**DOI:** 10.3390/e21070660

**Published:** 2019-07-05

**Authors:** Sergei Koltcov, Vera Ignatenko, Olessia Koltsova

**Affiliations:** St. Petersburg School of Physics, Mathematics, and Computer Science, National Research University Higher School of Economics, Kantemirovskaya Ulitsa, 3A, St. Petersburg 194100, Russia

**Keywords:** Sharma–Mittal entropy, topic modeling, optimal number of topics, stability

## Abstract

Topic modeling is a popular approach for clustering text documents. However, current tools have a number of unsolved problems such as instability and a lack of criteria for selecting the values of model parameters. In this work, we propose a method to solve partially the problems of optimizing model parameters, simultaneously accounting for semantic stability. Our method is inspired by the concepts from statistical physics and is based on Sharma–Mittal entropy. We test our approach on two models: probabilistic Latent Semantic Analysis (pLSA) and Latent Dirichlet Allocation (LDA) with Gibbs sampling, and on two datasets in different languages. We compare our approach against a number of standard metrics, each of which is able to account for just one of the parameters of our interest. We demonstrate that Sharma–Mittal entropy is a convenient tool for selecting both the number of topics and the values of hyper-parameters, simultaneously controlling for semantic stability, which none of the existing metrics can do. Furthermore, we show that concepts from statistical physics can be used to contribute to theory construction for machine learning, a rapidly-developing sphere that currently lacks a consistent theoretical ground.

## 1. Introduction

The Internet and, particularly, social networks generate a huge amount of data of different types (such as images, texts, or table data). A large amount of collected data becomes comparable to physical mesoscopic systems. Correspondingly, it becomes possible to use machine learning methods based on methods of statistical physics to analyze such data. Topic Modeling (TM) is a popular machine learning approach to soft clustering of textual or visual data, the purpose of which is to define the set of hidden distributions in texts or images and to sort the data according to these distributions. To date, a relatively large number of probabilistic topic models with different methods of determining hidden distributions have been developed, and several metrics for measuring the quality of topic modeling results have been formulated and investigated. The lion’s share of the research on TM has focused on the use of probabilistic models [1] such as variants of Latent Dirichlet Allocation (LDA) and probabilistic Latent Semantic Analysis (pLSA); therefore, we study and provide numerical experiments for these models. Non-probabilistic algorithms, such as Non-negative Matrix Factorization (NMF), can also be applied to the task of TM [2,3]; however, NMF approaches are less popular due to their inability to produce generative models. Other problems of NMF models were described in [4,5]. At the same time, despite broad usage of probabilistic topic models in different fields of machine learning [6,7,8,9], they, too, possess a set of problems limiting their usage for big data analysis.

A fundamental problem of probabilistic TM is finding the number of components in the mixture of distributions since the parameter determining this number has to be set explicitly [10,11,12]. A similar problem arises for the NMF approach since factorization rank has to be chosen [4]. A well-known exception is the Hierarchical Dirichlet Process model (HDP) [13] positioned by the authors as able to select the number of topics automatically. However, this class of models possesses a set of hidden parameters, which, according to the authors themselves, can influence the results of determining hidden distributions and the optimal number of topics correspondingly. The second unsolved problem in probabilistic TM is a certain level of semantic instability resulting from the ambiguity in retrieving the multidimensional density of the mixture of distributions. This ambiguity means that different runs of the algorithm on the same source data lead to different solutions. Solutions may differ both in terms of word and text composition of the resulting topics, which is usually incompatible with reliability requirements set by TM end users. The problems that have non-unique or non-stable solutions are termed ill-posed [14]. Let us mention that NMF is also an ill-posed problem [4] since factorization is not unique. A general approach to avoiding multiple solutions is given by Tikhonov regularization [14]. The essence of regularization is to redefine prior information that allows for narrowing the set of solutions. Regularization is implemented by introducing restrictions on hidden distributions [15], by modifying the sampling procedure [16], by using a combination of conjugate functions [10], or by incorporating different types of regularization procedures into the algorithm [15,17]. However, introduction of the regularization procedure, although it may contribute to higher stability, may also lead to the problem of determining regularization coefficients of probabilistic topic models since these parameters are, again, to be set by a user explicitly. All this leads users of machine learning methods to an understandable mistrust towards the obtained results [9,12].

The above problems naturally affect the quality of TM. Currently, the main methods for determining the quality of topic models are Gibbs–Shannon entropy [18,19], Kullback–Leibler divergence [20], log-likelihood [21], the Jaccard index [22,23], semantic coherence [17], and relevance [24]. However, first, each of these metrics measures only one of the aspects of TM performance. It is known that the distribution of words, at least in European languages, satisfies the so-called Zipf law (a power-law distribution), which is characteristic of complex systems, i.e., of systems with non-Markov processes [25,26]. It is known that the most effective way to investigate the behavior of complex systems is application of mathematical formalism borrowed from the theory of non-additive systems [26]. The goal of our research is thus to propose a metric that would be able to both measure different aspects of TM performance at the same time and would be more adequate for textual complex systems. For this, we adapt the mathematical formalism of non-extensive statistical physics, namely Sharma–Mittal entropy, and apply it for the analysis of the results of machine learning methods. We show that our metric combines the functionality of several existing metrics and is embedded in a more theoretically-grounded approach.

Before passing to our research, we briefly discuss the basics of TM and introduce notations. The key idea of TM is based on an assumption that any large document collection contains a set of topics or semantic clusters, while each word and each text of such a collection belongs to each topic with a certain probability. This gives TM an important ability to co-cluster both words by topics and topics by documents simultaneously. Topics are defined as hidden distributions of both words and texts that are to be restored from the observed co-occurrences of words in texts. Mathematically, topic models are based on the following propositions [27]:Let $\tilde{D}$ be a collection of textual documents with *D* documents and $\tilde{W}$ be a set (dictionary) of all unique words with *W* elements. Each document $d\in \tilde{D}$ is a sequence of terms $w_{1},\ldots ,w_{n}$ from dictionary $\tilde{W}$.It is assumed that there is a finite number of topics, *T*, and each entry of a word *w* in document *d* is associated with some topic $t\in \tilde{T}$. A topic is understood as a set of words that often (in the statistical sense) appear together in a large number of documents.A collection of documents is considered a random and independent sample of triples $\left(w_{i};d_{i};t_{i}\right)$, i=1,…,n, from the discrete distribution p(w;d;t) on a finite probability space $\tilde{W}\times \tilde{D}\times \tilde{T}$. Words *w* and documents *d* are observable variables, and topic *t* is a latent (hidden) variable.It is assumed that the order of words in documents is unimportant for topic identification (the ”bag of words” model). The order of documents in the collection is also not important.

In TM, it is also assumed that the probability p(w|d) of the occurrence of term *w* in document *d* can be expressed as a product of probabilities p(w|t) and p(t|d), where p(w|t) is the probability of word *w* under topic *t* and p(t|d) is the probability of topic *t* in document *d*. According to the formula of total probability and the hypothesis of conditional independence, one obtains the following expression [27]: $p\left(w|d\right)=\sum_{t\in \tilde{T}}p\left(w|t\right)p\left(t|d\right)\equiv \sum_{t\in \tilde{T}}\phi_{wt}\theta_{td}$. Thus, constructing a topic model means finding the set of latent topics $\tilde{T}$, i.e., the set of one-dimensional conditional distributions $p\left(w|t\right)\equiv \phi_{wt}$ for each topic *t*, which constitute matrix Φ (distribution of words by topics), and the set of one-dimensional distributions $p\left(t|d\right)\equiv \theta_{td}$ for each document *d*, which form matrix Θ (distribution of documents by topics), based on the observable variables *d* and *w*.

One can distinguish three types of models in the literature that allow solving this problem: (1) models based on likelihood maximization; (2) models based on Monte Carlo methods; and (3) models of the hierarchical Dirichlet process. A description of these models and their limitations can be found in Appendix A. In the process of TM, for algorithms based on the Expectation-Maximization (E-M) algorithm (first type) and the Gibbs sampling algorithm (second type), transition to a strongly non-equilibrium state occurs. The initial distributions of words and documents in matrices Φ and Θ for Gibbs sampling methods are flat; however, in E-M models, the initial distribution is determined by a random number generator. For both types of algorithm, the initial distribution corresponds to the maximum entropy of the topic model. Regardless of the algorithm type and the procedure of initialization, redistribution of words and documents by topics proceeds so that a significant portion of words (about 95% of all unique words) acquires probabilities close to zero and only about 3–5% receive probabilities above a threshold 1/W [28]. Numerical experiments demonstrate that the number of words with high probabilities depends on the number of topics and values of model parameters, which allows constructing a theoretical approach for analyzing such dependency using the perspective of statistical physics [29].

The rest of the paper proceeds as follows. Section 2.1 reviews the standard metrics, which are used in the field of machine learning, relationships between these metrics and their differences. Section 2.2 describes our concept and basic assertions of our new method. Section 2.3 is devoted to the adaptation of Renyi entropy for the analysis of TM results. Section 2.4 and Section 2.5 represent adaptation of Sharma–Mittal entropy for the analysis of TM results leading to a new quality metric in the field of TM. The relations of this new metric to the standard ones are also presented throughout Section 2. Section 3 shows numerical results of the application of our new metric to the analysis of TM outputs. We demonstrate the results of simulations run on two datasets by using two TM algorithms, namely probabilistic Latent Semantic Analysis (pLSA) and Latent Dirichlet Allocation (LDA) with Gibbs sampling. In Section 3, we also demonstrate the application of several standard metrics to TM results and compare them with our new metric. Section 4 summarizes the functionality of our two-parametric entropy approach and proposes directions for future research. Appendix A contains a short discussion of topic models and a detailed description of the models that were used in numerical experiments. Appendix B contains numerical results on another metric, which is called “semantic coherence”, to the outputs of TM and demonstrates difficulties when using this metric for tuning model parameters.

## 2. Materials and Methods

### 2.1. Methods for Analyzing the Results of Topic Modeling

The results of TM depend on the parameters of models, such as “number of topics”, hyper-parameters of Dirichlet distributions, or regularization coefficients, since these parameters are included explicitly in the mathematical formulation of the model. In the literature on TM, the most frequently-used metrics for analyzing topic models are the following.

Shannon entropy and relative entropy. Shannon entropy is defined according to the following equation [19,30,31]: $H=-\sum_{i=1}^{n}p\left(x_{i}\right)\log\left(p\left(x_{i}\right)\right)$, where $p(x_{i})$, i=1,…,n are distribution probabilities of a discrete random value with possible values $\left\{x_{1},\ldots ,x_{n}\right\}$. Relative entropy is defined as follows [32]: $D_{KL}\left(p|q\right)=\sum_{i}p\left(x_{i}\right)\log\left(\frac{p(x_{i})}{q(x_{i})}\right)=-\sum_{i}p\left(x_{i}\right)\log\left(q\left(x_{i}\right)\right)+\sum_{i}p\left(x_{i}\right)\log\left(p\left(x_{i}\right)\right)$, i.e., $D_{KL}\left(p|q\right)$ is the difference of cross-entropy $H\left(p,q\right)=-\sum_{i}p\left(x_{i}\right)\log\left(q\left(x_{i}\right)\right)$ and Shannon entropy. Relative entropy is also known as Kullback–Leibler (KL) divergence. In the field of statistical physics, it was demonstrated that KL divergence is closely related to free energy. In the work [33], it was shown that in the framework of Boltzmann–Gibbs statistics, KL divergence can be expressed as follows: $D_{KL}\left(p|\tilde{p}\right)=q\left(F\left(p\right)-F\left(\tilde{p}\right)\right)$, where *p* is the probability distribution of the system residing in the non-equilibrium state, $\tilde{p}$ is the probability distribution of the system residing in the equilibrium state, q=1/T, *T* is the temperature of the system, and *F* is the free energy. Hence, KL divergence is nothing but the difference between the free energies of off-equilibrium and equilibrium. The difference between free energies is a key characteristic of the entropy approach [29], which is to be discussed further below in Section 2.2 and Section 2.3. The variant of KL divergence used in TM is also discussed in Paragraph 3 of this section.Log-likelihood and perplexity: One of the most-used metrics in TM is the log-likelihood, which can be expressed through matrices Φ and Θ in the following way [21,34]: $\ln\left(P\left(\tilde{D}|\Phi ,\Theta \right)\right)=\sum_{d=1}^{D}\sum_{w=1}^{W}n_{dw}\ln\left(\sum_{t=1}^{T}\phi_{wt}\theta_{td}\right)$, where $n_{dw}$ is the frequency of word *w* in document *d*. A better model will yield higher probabilities of documents, on average [21]. In addition, we would like to mention that the procedure of log-likelihood maximization is a special case of minimizing Kullback–Leibler divergence [35]. Another widely-used metric in machine learning, and in TM, particularly, is called perplexity. This metric is related to likelihood and is expressed as: $\mathrm{perplexity}=\exp(-\ln\left(P\left(\tilde{D}|\Phi ,\Theta \right)\right)/\sum_{d=1}^{D}n_{d})$, where $n_{d}$ is the number of words in document *d*. Perplexity behaves as a monotone decreasing function [36]. The score of perplexity is the lower the better. In general, perplexity can be expressed in terms of cross-entropy as follows: $\mathrm{perplexity}=2^{\mathrm{entropy}}$ or $\mathrm{perplexity}=e^{\mathrm{entropy}}$ [37], where “entropy” is cross-entropy. The application of perplexity for selecting values of model parameters was discussed in many papers [10,17,21,34,38,39]. In a number of works, it was demonstrated that perplexity behaves as a monotonously-decreasing function of the number of iterations, which is why perplexity has been proposed as a convenient metric for determining the optimal number of iterations in TM [11]. In addition, the authors of [12] used perplexity for searching the optimal number of topics. However, the use of perplexity and log-likelihood has some limitations, which were demonstrated in [40]. The authors showed that perplexity depends on the size of vocabulary of the collection for which TM is implemented. The dependence of the perplexity value on the type of topic model and the size of the vocabulary was also demonstrated in [41]. Hence, comparison of topic models for different datasets and in different languages by means of perplexity is complicated. Many numerical experiments described in the literature demonstrate monotone behavior of perplexity as a function of the number of topics. Unlike the task of determining the number of iterations, the task of finding the number of topics is sensitive to this feature, and fulfillment of the latter task appears to be complicated by it. In addition, calculation of perplexity and log-likelihood is extremely time consuming, especially for large text collections.Kullback–Leibler divergence: Another measure, that is frequently used in machine learning, is the Kullback–Leibler divergence (KL) or relative entropy [32,42,43]. However, in the field of TM, symmetric KL divergence is most commonly used. This measure was proposed by Steyvers and Griffiths [20] for determining the number of stable topics: $KL\left(i,j\right)=\frac{1}{2}\sum_{w=1}^{W}\phi_{wi}^{\prime }\log_{2}\left(\frac{\phi_{wi}^{\prime }}{\phi_{wj}^{\prime \prime }}\right)+\frac{1}{2}\sum_{w=1}^{W}\phi_{wj}^{\prime \prime }\log_{2}\left(\frac{\phi_{wj}^{\prime \prime }}{\phi_{wi}^{\prime }}\right)$, where $\phi^{\prime }$ and $\phi^{\prime \prime }$ correspond to topic-word distributions from two different runs; *i* and *j* are topics. Therefore, this metric measures dissimilarity between topics *i* and *j*. Let us note that KL divergence is calculated for the same words in different topics; thus, the semantic component of topic models is taken into account. This metric can be represented as a matrix of size T·T, where *T* is the number of topics in compared topic models. The minimum of KL(i,j) characterizes the measure of similarity between topics *i* and *j*. If KL(i,j)≈0, then topics *i* and *j* are semantically identical. An algorithm for searching for the number of stable topics for different topic models was implemented [17] based on this measure. In this approach, pair-wise comparison for all topics of one topic’s solution with all topics of another topic solution was done. Hence, if the topic is stable from the semantic point of view, then it reproduces regularly for each run of TM. In [16], it was shown that different types of regularization lead to different numbers of stable topics for the same dataset. The disadvantage of this method is that this metric does not allow comparing one topic solution with another as a whole, but one can only obtain a set of pair-wise compared word distributions for separate topics. No generalization of this metric for solution-level comparisons has been offered yet.The Jaccard index and entropy distance: Another widely-used metric in the field of machine learning is the Jaccard index, also known as the Jaccard similarity coefficient, which is used for comparing the similarity and diversity of sample sets. The Jaccard coefficient is defined as the cardinality of the intersection of the sample sets divided by the cardinality of the union of the sample sets [23]. Mathematically, it is expressed as follows. Assume that we have two sets *X* and *Y*. Then, one can calculate the following values: *a* is the number of elements of *X*, which are absent in *Y*; *b* is the number of elements of *Y*, which are absent in *X*; *c* is the number of common elements of *X* and *Y*. The Jaccard coefficient is $J=\frac{c}{a+b+c}$, where c=|X∩Y|, |X∪Y|=a+b+c, |·| is the cardinality of a set. The Jaccard coefficient J=1 if sets are totally similar and J=0 if sets are totally different. This coefficient is used in machine learning due to the following reasons. Kullback–Leibler divergence characterizes similarity based on the probability distribution. This means that two topics are similar if words’ distributions for them have similar values. At the same time, the Jaccard coefficient demonstrates the number of identical words in topics, i.e., it reflects another point of view of the similarity of topics. The combination of two similarity measures allows for deeper analysis of TM results. In addition, the Jaccard distance is often used, which is defined as [22]: $J\left(X,Y\right)=1-\frac{c}{a+b+c}$. This distance equals zero if sets are identical. The Jaccard distance also plays an important role in computer science, especially, in research on “regular language” [44,45] and is related to entropy distance as follows [22]: $D_{H}\left(X,Y\right)=1-I\left(X,Y\right)/H\left(X,Y\right)=J\left(X,Y\right)=1-J$, where $D_{H}\left(X,Y\right)$ is entropy distance, I(X,Y) is the mutual information of *X* and *Y*, and H(X,Y) is the joint entropy of *X* and *Y*. In the standard set-theoretic interpretation of information theory, the mutual information corresponds to the intersection of sets *X* and *Y* and the joint entropy to the union of *X* and *Y*, and hence, the entropy distance corresponds to the Jaccard distance [22]. Correspondingly, if J(X;Y)=0, then $D_{H}\left(X,Y\right)=0$ as well. The paper proposes to use the Jaccard coefficient as a parameter of entropy, but not for TM tasks, while we incorporate it into our two-parametric entropy approach to TM specifically.Semantic coherence: This metric was proposed to measure the interpretability of topics and was demonstrated to correspond to human coherence judgments [17]. Topic coherence can be calculated as follows [17]: $C\left(t,W\left(t\right)\right)=\sum_{m=2}^{M}\sum_{l=1}^{m-1}\log\left(\frac{D(v_{m}^{t},v_{l}^{t})+1}{D(v_{l}^{t})}\right),$ where $W\left(t\right)=\left(v_{1}^{t},\ldots ,v_{M}^{t}\right)$ is a list of *M* most probable words in topic *t*, D(v) is the number of documents containing word *v*, and $D(v,v^{\prime })$ is the number of documents where words *v* and $v^{\prime }$ co-occur. The authors of [17] proposed to consider the following values of M=5,…,20. To obtain a single coherence score of a topic solution, one needs to aggregate obtained individual topic coherence values. In the literature, one can find that aggregation can be implemented by means of the arithmetic mean, median, geometric mean, harmonic mean, quadratic mean, minimum, and maximum [46]. Coherence can also be used for determining the optimal number of topics; however, in paper [47], it was demonstrated that the coherence score monotonously decreases if the number of topics increases.Relevance: This is a measure that allows users of TM to rank terms in the order of their usefulness for topic interpretation [24]. This measure is similar to a measure proposed in [48], where a term’s frequency is combined with the exclusivity of the word (exclusivity is the degree to which a word’s occurrences are limited to only a few topics). The relevance of term *w* to topic *t* given a weight parameter λ (0≤λ≤1) can be expressed as: $r\left(w,k|\lambda \right)=\lambda \cdot \log\left(\phi_{wt}\right)+\left(1-\lambda \right)\log\left(\frac{\phi_{wt}}{p_{w}}\right)$, where λ determines the weight given to $\phi_{wt}$ relative to its lift and $p_{w}$ is the empirical term probability, which can be calculated as: $p_{w}=\frac{\sum_{d=1}^{D}n_{dw}}{\sum_{d=1}^{D}n_{d}}$ with $n_{dw}$ being a count of how many times the term *w* appears in document *d* and $n_{d}$ being total term-count in document *d*, namely, $n_{d}=\sum_{w}n_{dw}$. The authors of [24] proposed to take the default value of λ=0.6 according to their user study; however, in general, it is not clear how to chose the optimal value of λ for a particular dataset. Furthermore, relevance is a topic-level measure that cannot be generalized for an entire solution, which is why it is not used further in this research.

### 2.2. Minimum Cross-Entropy Principles in Topic Modeling

As was shown above, TM parameter estimation and assessment of semantic stability are separate processes based on several unrelated metrics. Therefore, it is necessary to develop a single approach that would include a number of metrics and would allow solving simultaneously two problems, namely optimization of both semantic stability and other parameters. Such an approach can be developed on the basis of the cross-entropy minimum principle (minimum of KL divergence). In doing so, this principle can be implemented in two ways: (1) by constructing an entropic metric and searching for the minimum of this metric under variation of different topic model parameters, where TM is conducted using standard algorithms; (2) by creating an algorithm of restoring hidden distributions based on cross-entropy minimization. A version of the TM algorithm, close to the second approach, was considered in [49], where symmetric KL divergence was added to the model based on log-likelihood maximization. However, this model included regularization using only matrix Θ, and one has to set explicitly the regularization coefficient (the parameter called η). In our work, we consider only the first approach, i.e., searching for optimal parameters of the topic model based on the entropy metric, which takes into account the distribution of words by topics and the semantic stability of topics under the condition of the variation of different model parameters. By the “optimal” number of topics for a dataset, we mean the number of topics that corresponds to human judgment. We propose a method for tuning topic models, which is based on the following assertions [29,50], which create a linkage between TM and statistical physics and reformulate the problem of model parameter optimization in terms of thermodynamics: (1) A collection of documents is considered a mesoscopic information system: a statistical system where the elements are words and the documents number in the millions. Correspondingly, the behavior of such a system can be studied by application of models from statistical physics. (2) The total number of words and documents in the information system under consideration is constant (i.e., the system volume is not changed). (3) A topic is a state (an analogue of spin direction) that each word and document in the collection can take. Here, a word and a document can belong to different topics (spin states) with different probabilities. (4) A solution of topic modeling is a non-equilibrium state of the system. (5) Such information system is open and exchanges energy with the environment via changing the temperature. Here, the temperature of the information system is the number of topics that is a parameter and should be selected by searching for a minimum KL divergence. (6) Since KL divergence is proportional to the difference of free energies, to measure the degree to which a given system is non-equilibrium, one can use the following expression: $\Lambda_{F}=F\left(T\right)-F_{0}$, where $F_{0}$ is the free energy of the initial state (chaos) of the topic model and F(T) is the free energy after TM for a fixed number of topics *T* [50]. (7) The minimum of $\Lambda_{F}$ depends on topic model parameters such as the number of topics and other hyper-parameters. (8) The optimal number of topics and the set of optimal hyper-parameters of the topic model correspond to the situation when the information maximum (in terms of non-classical entropy) is reached. If one does not take semantic stability into account, then the information maximum corresponds to the Renyi entropy minimum [29]. However, in our work, we aim to consider the semantic stability of topics; hence, the information maximum will depend on the semantic component.

It is known that in topic models, the sum of probabilities of all words equals the number of topics $T=\sum_{t=1}^{T}\sum_{w=1}^{W}p_{wt}$, where $p_{wt}\in \left[0,1\right]$ for all w=1,…,W; t=1,…,T. In the framework of statistical physics, it is common to investigate the distribution of statistical systems by energy levels, where energy is expressed in terms of probability. In accordance with such approach, we divide the range of probabilities [0,1] by a fixed number of intervals, determine energy levels corresponding to these intervals, and then seek the number of words belonging to each energy level. Let us note that these values depend on the number of topics and the values of the hyper-parameters of a topic model. Division into intervals is convenient from a computational point of view. If the lengths of such intervals tend to zero, the distribution of words by intervals will tend to the probability density function. However, for simplification, we will consider a two-level system, where the first level corresponds to words with high probabilities and the second level corresponds to words with small probabilities close to zero. Therefore, we introduce the density-of-states function for words with high probabilities under a fixed number of topics and a fixed set of parameters: ρ=N/(WT), where *N* is the number of words with high probabilities. By high probability, we mean the probability satisfying: p>1/W. The choice of such a level is informed by the fact that the values 1/W are the initial values of matrix Φ for a topic model. The value W·T determines the total number of micro-states of the topic model (the size of matrix Φ), and normalizes the density-of-states function. During the process of TM, the probabilities of words redistribute with respect to the above threshold 1/W. A small part of the words has probabilities higher than the threshold level, while the larger part of words has probabilities lower than that. The energy of the upper level containing states with high probabilities is expressed as follows:(1)$$E=-\ln\left(\tilde{P}\right)=-\ln\left(\frac{1}{T}\sum_{wt}\left(p_{wt}\cdot \Omega \left(p_{wt}-1/W\right)\right)\right),$$
where the step function Ω(·) is defined by $\Omega (p_{wt}-1/W)=1$ if $p_{wt}\geq 1/W$ and $\Omega (p_{wt}-1/W)=0$ if $p_{wt}<1/W$. Therefore, in Equation (Equation 1), we sum only the probabilities that are greater than 1/W. The energy of the lower level is expressed analogously, except that summing occurs for probabilities that are smaller than 1/W. A level is characterized by two parameters: (1) the normalized sum of probabilities of micro-states, that lie in the corresponding interval, $\tilde{P}$; (2) the normalized number of micro-states (density-of-states function), ρ, whose probabilities lie in this interval. Let us note that the density-of-states function is sometimes called the statistical weight of a complex system’s level. For a two-level system, the main contribution to the entropy and energy of the whole system is made by the states with high probabilities, that is mainly by the upper level. Respectively, the free energy of the whole system is almost entirely determined by the entropy and the energy of the upper level. The free energy of a statistical system can be expressed through Gibbs–Shannon entropy and the internal energy in the following way [51]: F=E−TS=E−S/q, where q=1/T. The entropy of such a system can be expressed through the number of micro-states belonging to the same level [52]: S=ln(N). It follows that the difference of free energies of the topic model is expressed through $\tilde{P}$ and ρ in the following way:(2)$$\begin{array}{cc} \Lambda_{F}= & F\left(T\right)-F_{0}=\left(E\left(T\right)-E_{0}\right)-\left(S\left(T\right)-S_{0}\right)T\\ = & -\ln\left(\tilde{P}\right)-T\ln\left(\rho \right), \end{array}$$
where $E_{0}$ and $S_{0}$ are the energy and the entropy of the initial state of the system, with $E_{0}=-\ln\left(T\right)$ and $S_{0}=\ln\left(WT\right)$. Hence, the degree to which a given system is non-equilibrium can be defined as the difference between the two free energies and expressed in terms of experimentally-determined values ρ and $\tilde{P}$. Values ρ and $\tilde{P}$ were calculated for each topic model under variation of parameter *T* and hyper-parameters, i.e., $\Lambda_{F}$ is a function of the number of topics *T*, hyper-parameters, and size of vocabulary *W*.

### 2.3. Renyi Entropy of the Topic Model

Using partition function:(3)$$Z_{q}=e^{-q\Lambda_{F}}=e^{-qE+S}=\rho {\left(\tilde{P}\right)}^{q},$$

q=1/T [53], one can express Renyi entropy in Beck notation through free energy [54] and through experimentally-determined values ρ and $\tilde{P}$:(4)$$S_{q}^{R}=\frac{\ln(Z_{q})}{q-1}=\frac{\ln(e^{-q\Lambda_{F}})}{q-1}=\frac{-q\Lambda_{F}}{q-1}=\frac{q\ln\left(\tilde{P}\right)+\ln\left(\rho \right)}{q-1},$$
where, again, q=1/T. The choice of entropy in Beck notation is determined by the following considerations. Firstly, constructing topic models with just one or two topics is meaningless in terms of their informativeness for end users. Correspondingly, the entropy of such a model should be large. Secondly, excessive increase of the number of topics leads to a flat distribution of words by topics that, again, should lead to a large value of entropy. Thirdly, both *q* and $Z_{q}$ calculated for words with high probabilities are less than one. Correspondingly, if we normalize this value by 1−q, we will obtain a negative value of Renyi entropy. Taking into account the necessity to have maximum entropies at the boundaries of the range of the number of topics, the normalization coefficient q−1 should be used. Summing up the advantages of Renyi entropy application to TM, the following can be said. First, since calculation of Renyi entropy is based on the difference of free energies (i.e., on KL divergence or relative entropy), it is convenient to use Renyi entropy as a measure of the degree to which a given system is in non-equilibrium, and this is what we do in our approach. Second, Renyi entropy, in contrast to Gibbs–Shannon entropy, allows taking into account two different processes: a decrease in Gibbs–Shannon entropy and an increase in internal energy, both of which occur with the growth of the number of topics. The difference between these two processes can have an area of balance when two processes counterbalance each other. In this area, Renyi entropy reaches its minimum. Third, the search for the Renyi entropy minimum (i.e., minimum of KL divergence) can be convenient for optimizing regularization coefficients in topic modeling. As mentioned above, a relative drawback of Renyi entropy here is the impossibility of taking into account the semantic component of topic models since it is expressed only through the density-of-states function and energy of the level. However, this drawback can be overcome by using two-parametric Sharma–Mittal entropy, where one of deformation parameters is taken as q=1/T and the second deformation parameter corresponds to the semantic component of a topic model.

### 2.4. Sharma–Mittal Entropy in Topic Modeling

Sharma–Mittal two-parametric entropy proposed in [55] has been discussed in many works [56,57,58]. The main emphasis in these papers was made on the investigation of its mathematical properties [56,59,60] or application of this entropy when constructing generalized non-extensive thermodynamics [61]. In the field of machine learning, Sharma–Mittal entropy is used in a few works, for instance, in [62]. Two-parametric Sharma–Mittal entropy can be written as:(5)$$S_{SM}=\frac{{\left(\sum_{i}p_{i}^{q}\right)}^{\left(1-r)/(q-1\right)}-1}{1-r},$$
where *r* and *q* are deformation parameters. The essence of deformation parameters *r* and *q* for TM can be determined based on consideration of limit cases. One can show that $\lim_{r\to 1}S_{SM}=S_{q}^{R}$ and $\lim_{r\to 0}S_{SM}=\exp\left(S_{q}^{R}\right)-1$. Since in TM, deformation parameter *q* can be defined through the number of topics (q=1/T), in order to use Sharma–Mittal entropy for the purposes of TM, one has to define the meaning of parameter *r*. Let us note that r∈[0;1] according to [55]. In addition, if r→1, then Sharma–Mittal entropy transforms into Renyi entropy; hence, in this case, the quality of topic model is defined only by Renyi entropy and deformation parameter *q*, i.e., by the number of topics. If r→0, then the value of entropy becomes large since $\lim_{r\to 0}S_{SM}=\exp\left(S_{q}^{R}\right)-1$. Based on the principle that maximum entropy corresponds to the information minimum, we conclude that the minimum value of parameter *r* corresponds to the minimum information and maximum entropy. Taking into account that entropy can be parameterized by the Jaccard coefficient and that semantic distance between two topic solutions can be estimated by entropy distance, we define *r* as a parameter being responsible for the semantic stability of the topic model under variation of the number of topics or hyper-parameters. Therefore, we define the value of *r* as equal to the value of the Jaccard coefficient (i.e., r:=J, where *J* is the Jaccard coefficient calculated for the sets of the most probable words for each pair of topic solutions). Consequently, $1-r=J(W^{\prime },W^{\prime \prime })$ is the entropy distance or Jaccard distance, where $W^{\prime }$ and $W^{\prime \prime }$ are the sets of the most probable words of the first topic solution and the second topic solution, correspondingly.

### 2.5. Sharma–Mittal Entropy for a Two-Level System

Based on Equations (Equation 4) and (Equation 5) and the statistical sum (Equation 3), the Sharma–Mittal entropy of the topic model in terms of experimentally-determined values ρ and $\tilde{P}$ can be defined as:(6)$$S_{SM}=\frac{Z_{q}^{\left(1-r)/(q-1\right)}-1}{1-r}=\frac{{\left({\tilde{P}}^{q}\rho \right)}^{\left(1-r)/(q-1\right)}-1}{1-r}.$$

On the one hand, application of Sharma–Mittal entropy allows estimating the optimal values of topic model parameters, such as hyper-parameters, and the number of topics, by means of searching for the minimum entropy, which, in turn, is characterized by the difference of entropies between the initial distribution and the distribution obtained after TM. On the other hand, it allows estimating the contribution of the semantic difference between any two topic solutions that, in turn, is influenced by values of hyper-parameters and the number of topics. Hence, the optimal values of topic model parameters correspond to the minimum Sharma–Mittal entropy, and the worst values of parameters correspond to the maximum entropy.

## 3. Results

### 3.1. Data and Computational Experiments

For our numerical experiments, the following datasets were used:Russian dataset (from the Lenta.ru news agency): a publicly-available set of 699,746 news articles in the Russian language dated between 1999 and 2018 from the Lenta.ru online news agency (available at [63]). Each news item was manually assigned to one of ten topic classes by the dataset provider. We considered a class-balanced subset of this dataset, which consisted of 8624 news texts (containing 23,297 unique words). It is available here at [64]. Below, we provide statistics on the number of documents with respect to categories (Table 1).Some of these topics are strongly correlated with each other. Therefore, the documents in this dataset can be represented by 7–10 topics.English dataset (the well-known “20 Newsgroups” dataset http://qwone.com/~jason/20Newsgroups/): 15,404 English news articles containing 50,948 unique words. Each of the news items belonged to one or more of 20 topic groups. Since some of these topics can be unified, 14–20 topics can represent the documents of this dataset [65]. This dataset is widely used to test machine learning models.

We conducted our numerical experiments using pLSA and LDA with Gibbs sampling. These models represent two different types of algorithms. The LDA model used here was based on the Gibbs sampling procedure, and the pLSA model was based on the E-M algorithm. A detailed description of these models can be found in Appendix A. Experiments on these models allowed us to estimate the usability of Sharma–Mittal entropy for two main types of algorithms. Topic modeling was conducted using the following software implementation: the package “BigARTM” (http://bigartm.org) was used for pLSA; GibbsLDA++ (http://gibbslda.sourceforge.net) for LDA (Gibbs sampling). All source codes were integrated into a single package “TopicMiner” (https://linis.hse.ru/en/soft-linis) as a set of dynamic link libraries. Each model was calculated under variation of the number of topics in the range of [2;50] in increments of one topic, and for LDA model, also values of hyper-parameters α and β were varied in the range of [0;1] in increments of 0.1 for each dataset. For each model and for each dataset, the following metrics were calculated: (1) log-likelihood; (2) Jaccard index; (3) Sharma–Mittal entropy; (4) semantic coherence.

#### 3.1.1. Results for the pLSA Model

The choice of pLSA model was determined by the fact that this model has only one parameter: the number of topics. Correspondingly, we can isolate the effect of this parameter on the values of the above metrics. Figure 1 plots the log-likelihood as a function of the number of topics for both datasets. One can see that increasing the number of topics led to a smooth increase of the log-likelihood. Thus, these curves did not allow determining the optimal number of topics due to the absence of any clear extrema. The difference between these two curves resulted from different sizes of vocabularies and the different amounts of documents in the corresponding datasets.

Figure 2 demonstrates Renyi entropy curves for the pLSA model on both datasets. The entropy was calculated according to Equation (Equation 4). The exact minimum of Renyi entropy for the Russian dataset was seven and for the English dataset 16. However, as was noted, being an ill-posed problem, topic modeling produced different results on different runs of the same algorithm, which was especially true for pLSA. From the previous research [29], it is known that the range of such variation between the runs is approximately ±3 topics. Therefore, it makes more sense to look at the range of the neighboring minima rather than at the exact minimum. It can be seen that the numbers of topics defined by humans, when corrected for inter-topic correlation, lied within the discovered ranges in both datasets, which suggests the language-independent character of this metric (at least for European languages). As Renyi entropy does not include an instrument to evaluate the semantic stability of topic models, we calculated Jaccard coefficients under variation of the number of topics. Figure 3 presents a “heat map” of Jaccard coefficients for the dataset in the Russian language. The matrix containing Jaccard coefficients was symmetric with respect to the main diagonal, and this is the reason why only half of this matrix is depicted. The structure of the “heat map” of the Jaccard index for the English dataset was similar to that for the Russian dataset and can be found in [29].

Figure 4 presents a pairwise comparison of topic solutions with the number of topics equal to *T* and T+1 correspondingly, under variation of *T* for the Russian and English datasets. As demonstrated in Figure 3 and Figure 4, there are areas of sharp decreases in semantic similarity between topic solutions with different numbers of topics. In order to incorporate the “density-of-states” function, the probabilities of words, and semantic similarity under variation of the parameter “number of topics”, we calculated Sharma–Mittal entropy according to Equation (Equation 6) for the pLSA model on both datasets.

Figure 5 plots Sharma–Mittal entropy as a function of the number of topics calculated only on the data from pairwise comparisons of topic solutions with the neighboring values of *T* (i.e., *T* and T+1). The values of the Jaccard index used for this calculation constitute over-diagonal elements taken from the full matrix of pairwise comparisons of all topic solutions in the range T=[2;50]. Figure 6 and Figure 7 demonstrate Sharma–Mittal entropy for the Russian and English datasets where large values (≥5) were replaced by five to make the global minimum more visible.

Figure 8 shows Sharma–Mittal entropy for the pLSA model in two versions: a 3D picture and its view from above. Together, they show that Sharma–Mittal entropy has areas of minima and maxima, the overall shape of the curve being determined by the number of topics and the local fluctuations resulting from the fluctuations of the Jaccard distance. In practice, however, we propose to consider only two-dimensional versions of this figure (e.g., Figure 6), where the Jaccard index is calculated only for the neighboring solutions. Such plots are easier to interpret, and at the same time, they demonstrate the influence of semantic stability. The exact values of the Sharma–Mittal entropy minimum are the following: T=20 for the English dataset and T=7 for the Russian dataset. Horizontal shift of the Sharma–Mittal entropy minimum as compared to the Renyi entropy minimum on the English dataset is an effect of the sharp fall of the Jaccard coefficient observed in the range of 14–16 topics. It follows that application of Sharma–Mittal entropy for models based on the E-M algorithm allows determining the optimal number of topics involving the semantic stability of topics. Figures that demonstrate the behavior of semantic coherence for these datasets can be found in Appendix B. We do not provide them here since they monotonously decrease, with some fluctuations, but without any clear extrema, thus providing no criteria for choosing topic number.

#### 3.1.2. Results for the LDA with Gibbs Sampling Model

The difference between the pLSA model and LDA Gibbs sampling model is not only in the application of the Monte Carlo algorithm for determining hidden distributions, but also in the presence of a regularization procedure. The level of regularization in LDA with Gibbs sampling is determined by hyper-parameters α and β. In our numerical experiments, we used the algorithm [11] where hyper-parameters of the LDA model were fixed and did not change from iteration to iteration since our goal was to analyze the results of the LDA model with respect to different values of hyper-parameters. Figure 9 plots the log-likelihood for the Russian dataset as a function of *T* for pLSA and for LDA with different fixed values of α or β. The behavior of the log-likelihood for the English dataset was similar to that for the Russian dataset, and therefore, we do not provide the figure.

Using the results of calculations (Figure 9), one can conclude that the log-likelihood metric allows estimating the effect of regularization in the LDA Gibbs sampling model. Namely, it can be seen that the largest values of regularization coefficients (blue curve) led to the lowest values of the log-likelihood, while according to [21,34], the optimal topic model should correspond to the maximum log-likelihood. According to our numerical results, the maximum log-likelihood corresponds to the pLSA model, that is to the zero regularization of LDA. Let us note that a similar result was obtained in [66], where, according to human mark-up, pLSA was shown to perform better than LDA, as regularized pLSA, and than pLSA regularized with decorrelation and sparsing-smoothing approaches, for the task of revealing ethnicity-related topics.

Figure 10 and Figure 11 plot Renyi entropy as functions of *T* for different values of α and β for the Russian and English datasets. Calculations demonstrated that application of Renyi entropy and the log-likelihood allowed estimating the influence of regularization in TM. Namely, larger regularization coefficients led to higher entropy, i.e., to the model’s deterioration. The exact minima of Renyi entropy were the following: (1) Russian dataset: T=7 for α=0.1,β=0.1; T=9 for α=0.5,β=0.1; T=14 for α=1,β=1; (2) English dataset: T=17 for α=0.1,β=0.1; T=15 for α=0.5,β=0.1; T=13 for α=1,β=1. It follows that Renyi entropy is useful for estimating topic model hyper-parameters for different datasets, at least in European languages. In addition, Renyi entropy is less sensitive to the size of vocabulary since this metric is normalized with respect to initial states (chaos). However, as Renyi entropy for the LDA Gibbs sampling model and pLSA model does not allow taking into account semantic stability, we further do not present our results on Sharma–Mittal entropy.

Figure 12 and Figure 13 show curves of Sharma–Mittal entropy for the LDA Gibbs sampling model under variation of hyper-parameters α and β for the Russian and English datasets. Figure 14 and Figure 15 demonstrate Sharma–Mittal entropy curves where large values (≥6) are replaced by six in order to demonstrate clearly the location of the global minimum. These figures show that for small values of hyper-parameters, the behavior of Sharma–Mittal entropy for LDA is similar to that for the pLSA model. The exact minima of Sharma–Mittal entropy were: (1) Russian dataset: T=7 for α=0.1,β=0.1; T=7 for α=0.5,β=0.1; T=19 for α=1,β=1; (2) English dataset: T=21 for α=0.1,β=0.1; T=21 for α=0.5,β=0.1; T=13 for α=1,β=1 Furthermore, these figures demonstrate that the location of jumps of Sharma–Mittal entropy, which are related to semantic stability, are almost independent of the regularization coefficients. However, in general, entropy curves were lifted along the *Y* axis if regularization coefficients increased. It follows that for LDA Gibbs sampling, the optimal values of both α and β coefficients were small. It can be concluded that the results of regularization coefficients’ selection by means of Sharma–Mittal entropy were similar to those obtained with the log-likelihood and Renyi entropy; however, two-parametric entropy, unlike other considered metrics, allowed incorporating semantic stability using the Jaccard distance. Sharma–Mittal entropy under variation of the number of topics and incorporation of the Jaccard coefficient represents a three-dimensional structure with a set of local minima, which are determined by the number of topics and by semantic stability. These areas of local minima represent islands of stability. Figure 16 and Figure 17 demonstrate the three-dimensional surfaces of Sharma–Mittal entropy for the Russian and English datasets and its projections to the horizontal plane $OT_{1}T_{2}$.

Numerical results on semantic coherence for LDA with Gibbs sampling can be found in Appendix B (Figure A3 and Figure A4). However, as with pLSA, this metric fell monotonously and did not provide any criteria for the choice of the topic number.

## 4. Discussion

In this work, we proposed a new entropy-based approach for the multi-aspect evaluation of the performance of topic models. Our approach was based on two-parametric Sharma–Mittal entropy, that is twice deformed entropy. We considered the deformation parameter, *q*, being the inverse value of the number of topics, and the second parameter, *r*, being the Jaccard coefficient, while 1−r the entropy distance. Our numerical experiments demonstrated that, firstly, Sharma–Mittal entropy, as well as Renyi entropy allowed determining the optimal number of topics. Secondly, as the minimum of Sharma–Mittal entropy corresponded to the maximum of the log-likelihood, the former also allowed choosing the optimal values of hyper-parameters. Thirdly, unlike Renyi entropy or the log-likelihood, it allowed optimizing both hyper-parameters and the number of topics, simultaneously accounting for semantic stability. This became possible due to the existence of areas of semantic stability that have been shown to be characterized by low values of Sharma–Mittal entropy. According to our numerical results, the location of such areas did not depend on the hyper-parameters. However, on the whole, larger values of hyper-parameters in the LDA Gibbs sampling led to higher entropy, while small values made the LDA model almost identical to pLSA. This means that new methods of regularization are needed that would not impair TM performance in terms of entropy. We concluded that Sharma–Mittal entropy is an effective metric for the assessment of topic models performance since it includes the functionality of several metrics.

However, our approach had certain limitations. First of all, topic models have an obvious drawback, which is expressed by the fact that the probabilities of words in topics depend on the number of documents containing these words. This means that if a topic is represented in a small number of documents, then the topic model will assign small probabilities to the words of this topic, and correspondingly, a user will not be able to see this topic. Thus, topic models can detect topics that are represented in many documents and poorly identify topics with a small number of documents. Therefore, Renyi entropy and Sharma–Mittal entropy allow determining the number of those large topics only. Secondly, in our work, Sharma–Mittal entropy was tested only for two European languages, while there are papers on the application of topic models for the Chinese, Japanese, and Arabic languages. Correspondingly, our research should be extended and tested on non-European languages. Thirdly, our metric allowed finding the global minimum when topic modeling was performed in a wide range of the number of topics; however, this process was resource-intensive and in practice can be applied to datasets containing up to 100–200 thousand documents. For huge datasets, this metric is not applicable. This problem might be partially solved by means of renormalization, which can be adapted for topic models from statistical physics. Research on application of renormalization for fast search of Renyi entropy and Sharma–Mittal entropy minima deserves a separate paper. Fourthly, we would like to note that our method was not embedded in algorithms of topic modeling. Therefore, in future research, the metric of quality based on Sharma–Mittal entropy can be used for the development of new topic models. Sharma–Mittal entropy can be embedded in the algorithms based on the Gibbs sampling procedure, where walks in the multi-dimensional space of words, hyper-parameters, and the number of topics will be determined by the level of this entropy. Correspondingly, transition along different axes of multi-dimensional space can be guided by the entropy minimization principle. An algorithm similar to the algorithm of annealing based on searching for the minimum Tsallis entropy [67] can be used in this case. However, unlike the algorithm proposed by Tsallis, one can use deformation parameter *q* as a parameter that controls the number of components in the mixture of distributions and search for the minimum when changing the number of components. Therefore, the walk in the multi-parameter space can be determined by the direction of the minimum of deformed entropy when changing the dimension of the space.

For topic models based on the maximum log-likelihood principle, the sizes of matrices are included in the model as external parameters, which are selected by the user. Correspondingly, new topic models can be developed in the future by using the principle of deformed logarithm maximization, where one of deformation parameters corresponds to the sizes of matrices (namely, the number of topics) and the other parameter corresponds to semantic stability (e.g., the Jaccard index). Note that both parameters here are maximization parameters. A more detailed discussion of these possible directions for research is out of the scope for this paper and can be used as a starting point for new research.

## Figures and Tables

**Figure 1 entropy-21-00660-f001:**
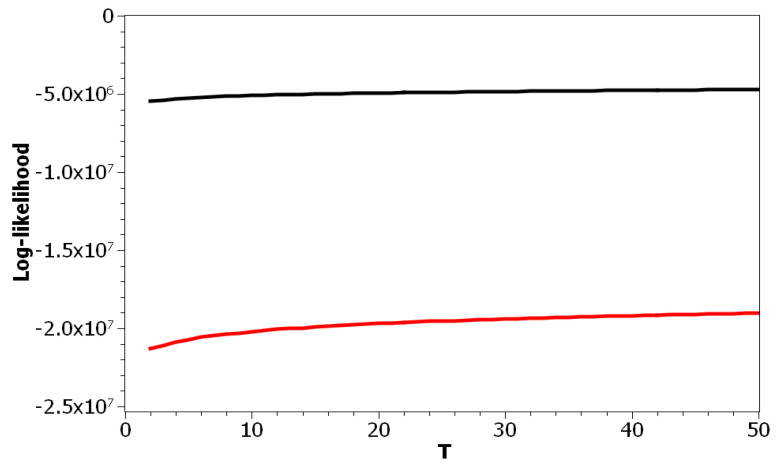
Log-likelihood distribution over *T* (probabilistic Latent Semantic Analysis (pLSA)). Russian dataset, black; English dataset, red.

**Figure 2 entropy-21-00660-f002:**
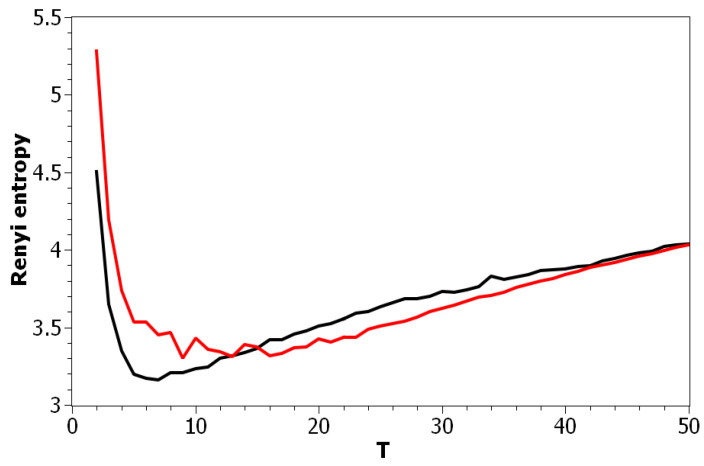
Renyi entropy distribution over the number of topics *T* (pLSA). Russian dataset, black; English dataset, red.

**Figure 3 entropy-21-00660-f003:**
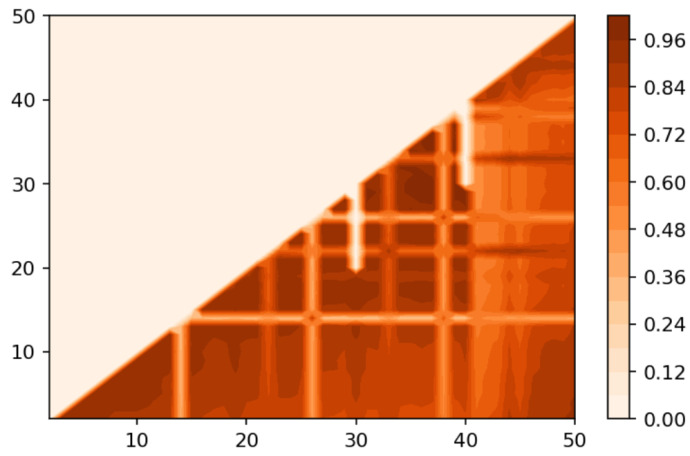
Heat map of the Jaccard index for the Russian dataset (pLSA).

**Figure 4 entropy-21-00660-f004:**
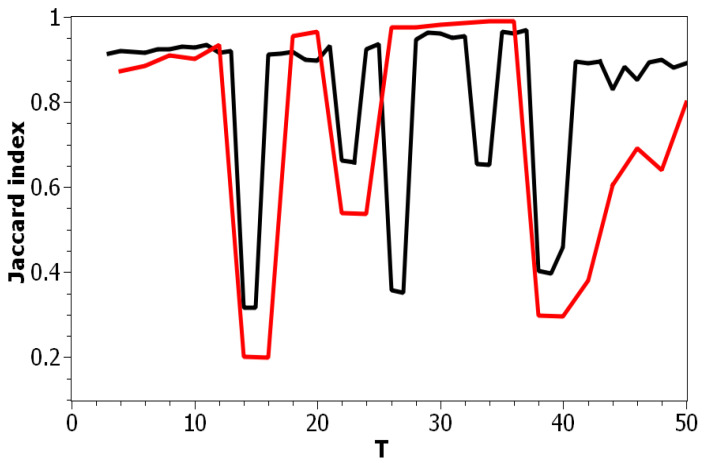
Distribution of Jaccard coefficients of the pairwise comparison for neighboring topic solutions with the number of topics *T* and T+1 (pLSA). Russian dataset, black; English dataset, red.

**Figure 5 entropy-21-00660-f005:**
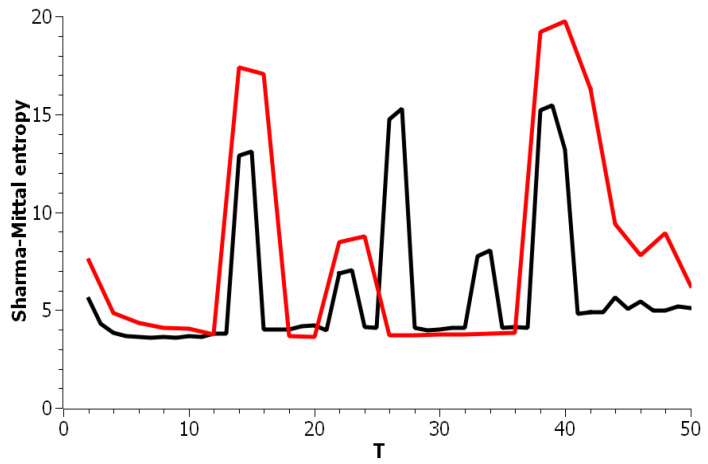
Sharma–Mittal entropy distribution over the number of topics *T* (pLSA). Russian dataset, black; English dataset, red.

**Figure 6 entropy-21-00660-f006:**
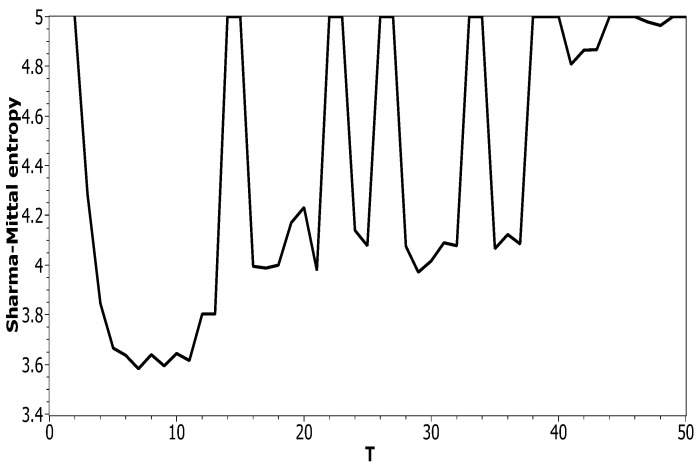
Sharma–Mittal entropy distribution over *T* with $S_{SM}$ > 5 reduced to five (Russian dataset). pLSA.

**Figure 7 entropy-21-00660-f007:**
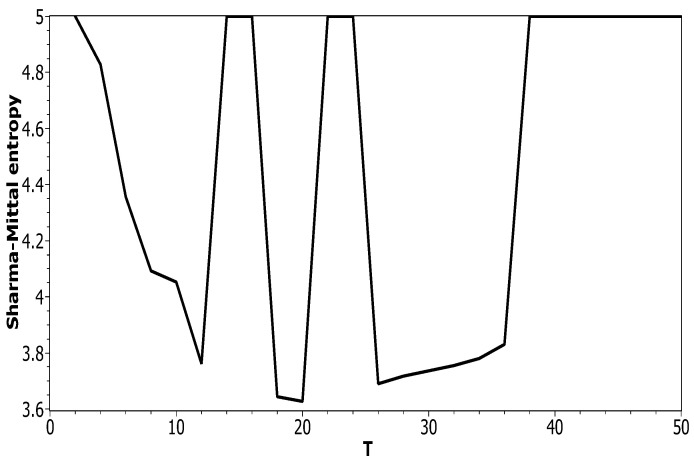
Sharma–Mittal entropy distribution over *T* with $S_{SM}$ > 5 reduced to five (English dataset). pLSA.

**Figure 8 entropy-21-00660-f008:**
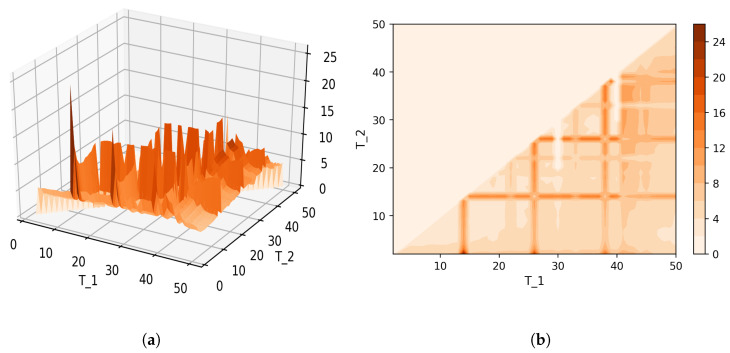
Sharma–Mittal entropy for the pLSA model (Russian dataset). (**a**) 3D plot of Sharma–Mittal entropy; (**b**) projection of Sharma–Mittal entropy to $OT_{1}T_{2}$.

**Figure 9 entropy-21-00660-f009:**
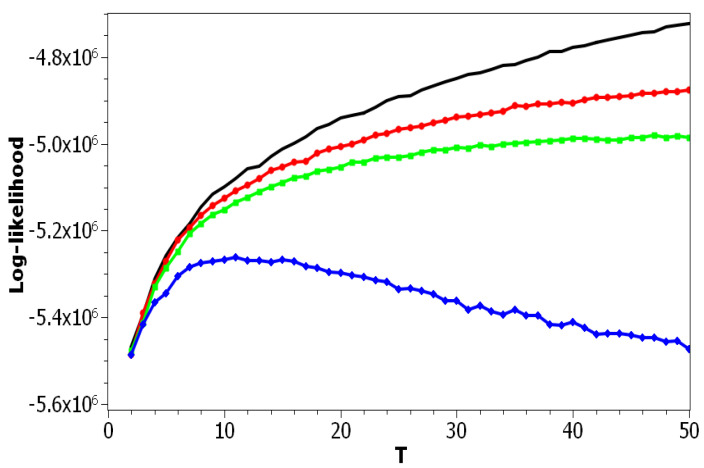
Log-likelihood distribution over *T* for different α and β (Russian dataset). pLSA, black; LDA (α = 0.1, β = 0.1), red; LDA (α = 0.5, β = 0.1), green; LDA (α = 1, β = 1), blue.

**Figure 10 entropy-21-00660-f010:**
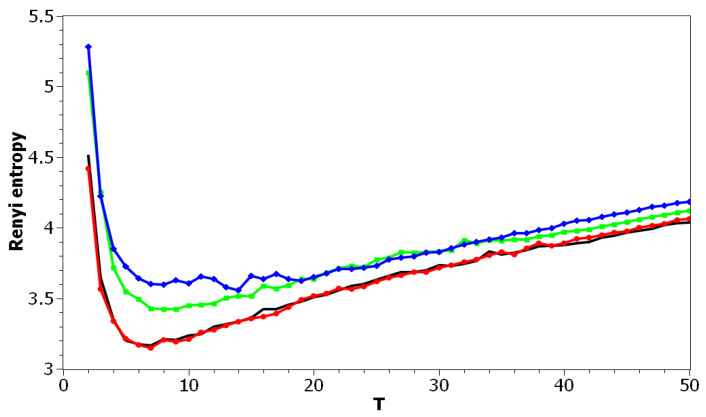
Renyi entropy distribution over *T* for different α and β (Russian dataset). pLSA—black, LDA (α = 0.1, β = 0.1)—red, LDA (α = 0.5, β = 0.1)—green, LDA (α = 1, β = 1)—blue.

**Figure 11 entropy-21-00660-f011:**
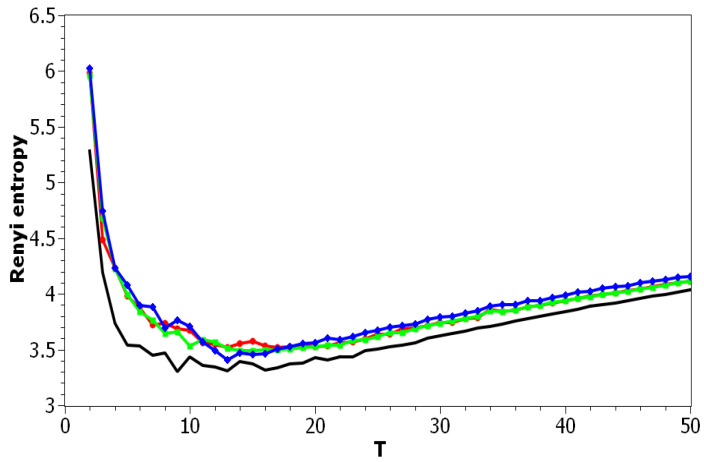
Renyi entropy distribution over *T* for different α and β (English dataset). pLSA, black; LDA (α = 0.1, β = 0.1), red; LDA (α = 0.5, β = 0.1), green; LDA (α = 1, β = 1), blue.

**Figure 12 entropy-21-00660-f012:**
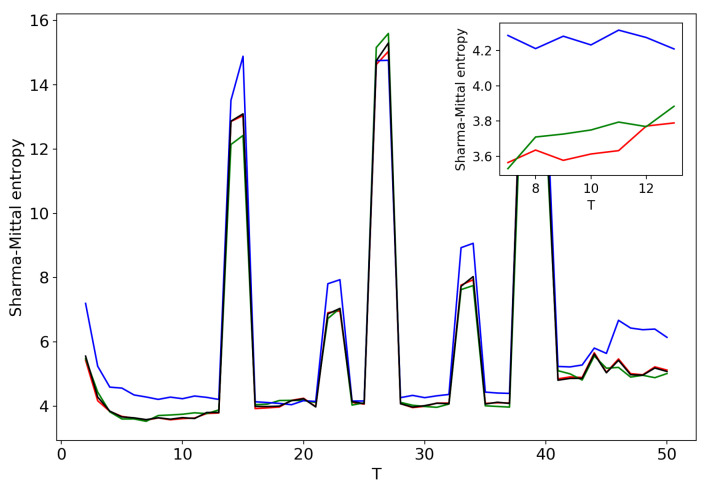
Sharma–Mittal entropy distribution over topics (Russian dataset). pLSA, black; LDA (α = 0.1, β=0.1), red; LDA (α=0.5, β=0.1), green; LDA (α=1, β=1), blue.

**Figure 13 entropy-21-00660-f013:**
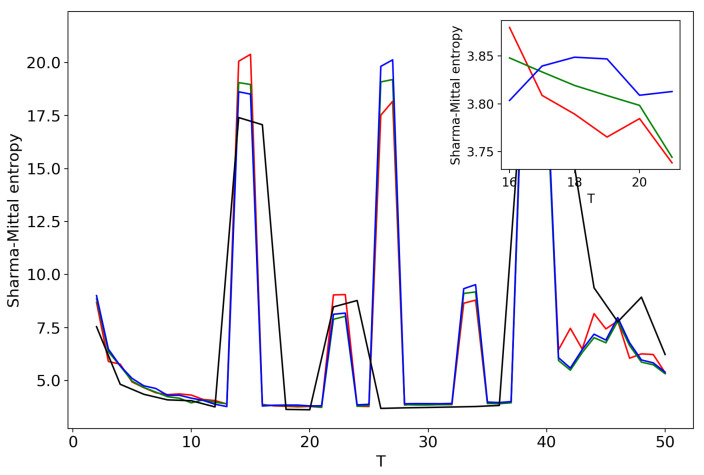
Sharma–Mittal entropy distribution over topics (English dataset). pLSA, black; LDA (α=0.1, β=0.1), red; LDA (α=0.5, β=0.1); green, LDA (α=1, β=1), blue.

**Figure 14 entropy-21-00660-f014:**
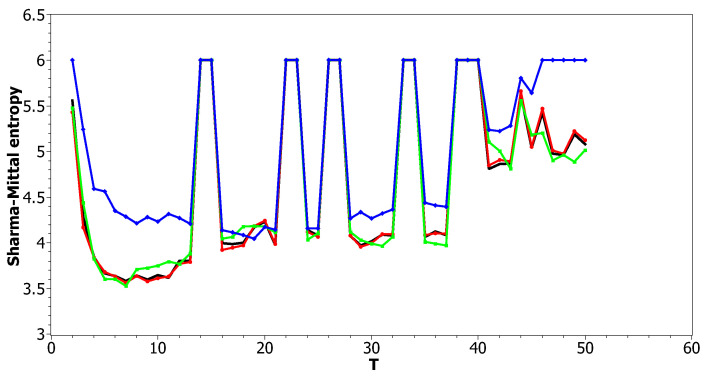
Sharma–Mittal entropy distribution over topics (Russian dataset). pLSA, black; LDA (α = 0.1, β=0.1), red; LDA (α=0.5, β=0.1), green; LDA (α=1, β=1), blue.

**Figure 15 entropy-21-00660-f015:**
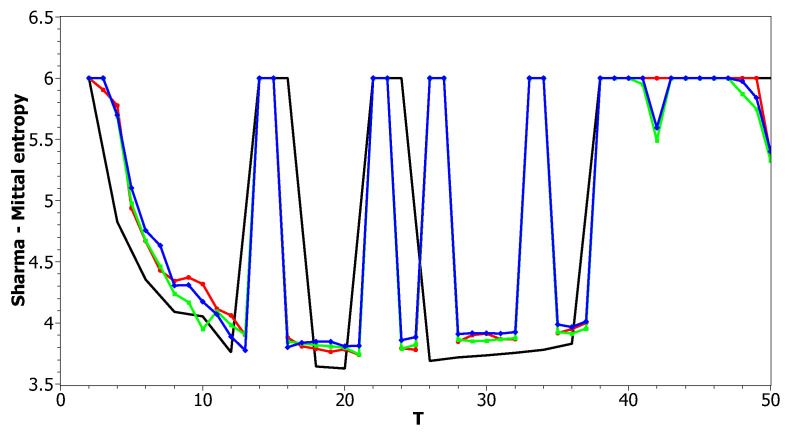
Sharma–Mittal entropy distribution over topics (English dataset). pLSA, black; LDA (α=0.1, β=0.1), red; LDA (α=0.5, β=0.1), green; LDA (α=1, β=1), blue.

**Figure 16 entropy-21-00660-f016:**
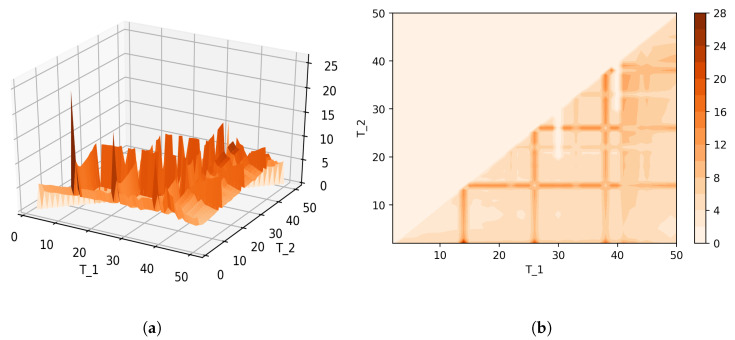
Sharma–Mittal entropy for the LDA model (Russian dataset). (**a**) 3D plot of Sharma–Mittal entropy; (**b**) projection of Sharma–Mittal entropy to $OT_{1}T_{2}$.

**Figure 17 entropy-21-00660-f017:**
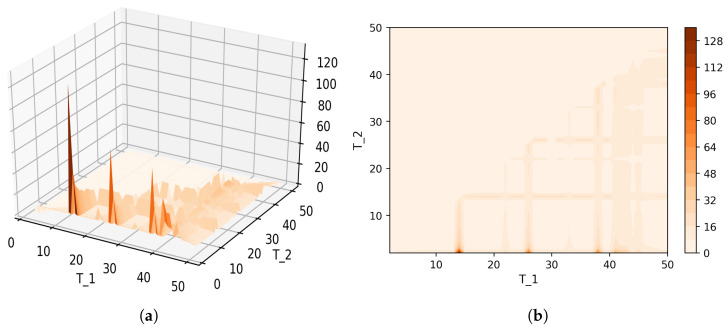
Sharma–Mittal entropy for the LDA model (English dataset). (**a**) 3D plot of Sharma–Mittal entropy; (**b**) projection of Sharma–Mittal entropy to $OT_{1}T_{2}$.

**Table 1 entropy-21-00660-t001:** Statistics on the Russian dataset.

Category	Number of Documents
business	466
culture	499
economy and finance	667
incidents	712
media	628
policy	1231
security services	863
science and tech	580
society and travel	1957
sports	1022

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
