# Peer review of "Estimating Topic Modeling Performance with Sharma–Mittal Entropy"

_entropy, 2019, doi:10.3390/e21070660_

Round 1
Reviewer 1 Report
This paper applies concepts from statistical physics to inform the perennial problem of topic selection, while also accounting for the semantic stability. Achieving solutions to these issues is extremely important to applied research on topic modelling and, as the authors note, drawing on concepts from statistical physics provides a useful lens through which to view (probabilistic) topic models. I do, however, have a number of comments that will hopefully improve the paper. I will provide these comments in the order in which they appear in the manuscript (not necessarily in order of importance):
1. The discussion of “Topic Models” in the introduction (and throughout the remainder of the text) is directed at probabilistic topic models in which problems of stability are well-known. However, you might want to draw a distinction between LDA-style models and those based on non-negative matrix factorization. They tend to suffer from different problems – i.e., stability concerns are generally less problematic for NMF models.
2. Your discussion of “frequently used metrics for analysing topic models” (Section 2.1) leaves out several of the most commonly used metrics in my view—e.g., so-called “semantic coherence” (or “salience”) and relevance (or “exclusivity”). It’s also more and more common to use these types of measures when trying to choose the “optimal” number of topics given well-known problems with perplexity (or held-out likelihood). You might want to engage with these measures as well, demonstrating the extent to which entropy-based measures offer advantages over coherence-based measures.
3. You say “[t]he optimal number of topics and the set of optimal hyper-parameters of topic model corresponds to situation when information maximum is reached.” This may be true. However, you have cited Chang et al. (2009) which clearly demonstrates that “optimal” from a human’s perspective is not always “optimal” from a statistical perspective. I would probably be a little more careful regarding the use of language here – i.e., be very clear what you mean be optimal. Moreover, I would really hammer home the extent to which your measures correspond to human judgement (you start to do this in your experiments, but I would be clearer)
4. On the numerical experiments:
a. I need more on how the manual annotation of the “Russian Dataset” – you provide very few details here (unless I missed something). There is no way that I could replicate this experiment (even in theory).
b. It would be great if you could add a couple of additional datasets here. This maybe outside the scope of this paper; however, I still have reservations of how well these results will carry outside of these datasets. Should these measures work equally well for corpora that are not newspapers? What about shorter texts?
c. On the LDA, in practice, the hyperparameters will almost always be optimised. Does this make any difference whatsoever for your findings? If not, it would be a good idea to say so.
5. You say “[f]urthermore, we show that concepts from statistical physics can be used to contribute to theory construction for machine learning – a rapidly developing sphere that currently lacks a consistent theoretical ground.” I think this is a really important point and potentially an important contribution. However, the “translation” between statistical physics and ML/NLP was not always clear throughout the manuscript. Coming from a background in NLP (and not statistical physics), I’m still not 100% sure where/when you offer important “theoretical insights” that will help scholars develop more informative topic models. Perhaps more can be said here.
Author Response
Dear reviewer, thank you for your valuable remarks. Please, find point-by-point replies to your comments. Your comments are in capital letters.
1. ONLY PROBABILISTIC TOPIC MODELS ARE DISCUSSED. To the best of our knowledge, practitioners usually prefer probabilistic topic models, therefore, we are focused on this type of models. We added several sentences concerning NMF models and their limitations in the text (lines 27-29, 35, 45).
2. ABSENCE OF SEMANTIC COHERENCE AND RELEVANCE METRICS. We included description of semantic coherence and relevance (lines 216-238). We also added numerical results on semantic coherence (Appendix B) and demonstrated that Sharma-Mittal entropy is more informative when choosing the number of topics for the datasets under consideration. Relevance is not computed as it is a topic-level metric that in principle cannot be generalized to the level of a topic solution (this has also been explained in the paper).
3. MEANING OF THE OPTIMAL NUMBER OF TOPICS. The ‘optimal’ number of topics is the number of topics which corresponds to human judgment; this definition has been introduced in the lines 254-255. Chang et al (2009) to whom you refer do not find a match between human judgement and statistical metrics, but our metric does match the human judgement. More importantly, Cnang et al do not question human judgement as the ultimate measurement of topic quality; instead, they call for the development of metrics more capable of simulating it, which is what we are trying to do. BE CLEAER ABOUT CORRESPONDENCE TO HUMAN JUDGEMENT. Clarifications are inserted into lines 419-420, 452, 486-487 and 499-500.
4. a. INSUFFICIENT INFO ON THE RUSSIAN DATASET. We provide the hyperlinks to the full Russian dataset and the subset we use in the lines 383, 386, as well as the information about its annotation. As the corpus was annotated by its providers, we only know that the annotation was single-class and contained 10 topics.
b. ADDITIONAL DATASETS. We created a separate file which is called Supplementary materials and is available with this link https://yadi.sk/i/bJgaqHSQpKgonw . This file contains numerical experiments on three additional datasets.
c. HYPER-PARAMETERS ARE USUALLY OPTIMISED IN LDA IN PRACTICE. To the best of knowledge, existing implementations of hyperparameter optimization are based on the work of Wallach et al whose work is, in turn, based on Minka’s approach. However, Wallach optimizes hyperparameters by maximizing log-likelihood and, unlike us, she does not test this approach on human mark-up. Log-likelihood, as we show, does not allow to combine optimization of hyperparameters with the optimization of topic number simultaneously accounting for sematic stability, which is what Sharma-Mittal entropy does. Currently, we are preparing a separate paper of hyperparameter optimization for topic models with different types of regularization, which is why we do not develop this line of argumentation in this paper.
5. TRANSLATION BETWEEN STATISTICAL PHYSICS AND ML/NLP, THEORETICAL INSIGHTS TO DEVELOP MORE INFORMATIVE TOPIC MODELS. The translation between statistical physics and topic modeling is given in section 2.2 (lines 256-257). Additionally, we have extended our Discussion section and believe that it will help scholars to develop new models with a strong background in statistical physics.
Reviewer 2 Report
The authors present a study of Topic Modeling based on
the application of Sharma-Mittal entropy.
The paper is well presented with a very complete Introduction to the problem. However, the results are not
strong, with rather some general discussions about the implications of the calculations.
My opinion is that the paper must be improved, specially in the results Section, in order to be suitable for publication.
Major Points:
1) Authors test their approac with the help of two models (pLSA and LDA), but a few explanations are given about the models. What are the physical/computational meaning of the parameters alpha and beta?.
3) According with the results, the behavior of the
Renyi entropy vs T (Fig. 2) exhibits a minimun value.
What are the specific values for English and Russian?.
What is the explanation of the crossing point between both entropies around T~14 or 16?. What is the value of q in Eq. 4 for this plot?
3) Figs. 4 and 5 present the results of Jaccard index and Sharma-Mittal entropy as a function of T, but these plots
seem to be very complicated to interpret. There are many
intervals with local minima and maxima, with clear tendency at all. What is your interpretation of such irregularity?. The minimum value identified in Fig. 6 is
a very risky election (line 391), what is the reason?
4) The results based on the calculations of Sharma-Mittal
entropy require to specify the values of q and r, what are the values ?.
5) The title of the paper indicates that the use of S-M entropy is quite important in your discussion, but it seems that S-M entropy renders very small variations or changes when tested with models and changes in the parameters. My opinion is that Renyi is quite more informative (prove me if I am wrong).
6) The Discussion must contain some sentences about the limitations of the study.
7) The title of this paper is quite similar to a previous one based on Tsallis and Renyi entropy. Please use another title.
Author Response
Dear reviewer, thank you for your valuable remarks. Please, find point-by-point replies to your comments.
1. NO DESCRIPTION OF PLSA AND LDA MODELS. We inserted a description of these models in Appendix A with a short explanation of the meaning of parameters alpha and beta (lines 692-696). In a nutshell, alpha and beta are the parameters of Dirichlet distribution that was initially proposed due to its mathematical convenience and by now has empirically shown better results than earlier models, e.g. pLSA.
2. NOT SPECIFIED VALUES OF ENTROPY MINIMUM. We inserted the specific values in lines 419-420, 452, 486-487 and 499-500.
NO EXPLANATION OF CROSSING OF ENTROPIES. It is just a coincidence and is not valuable. It happens because Renyi entropy does not depend on the size of vocabulary of the collection, therefore, its values lie in a certain region and can coincide for different datasets.
VALUE OF q IN EQ. 4. Here and everywhere q=1/T, where T is the number of topics (see line 321)
3. FIGS 4 & 5 VERY COMPLICATED FOR INTERPRETATION. We added figures (which is why their numbering has changed) which are easier for perception and added our recommendation for practitioners to consider two-dimensional figures instead of three-dimensional ones. So, the graphical representation has become more interpretable. We hope that new figures 6, 7, 13 and 15 allow the readers to see the tendency in Sharma-Mittal entropy behavior and to see clearly the global minimum.
MANY LOCAL EXTREMA WITH NO CLEAR TENDENCY - we introduced two new figures 6 and 7 that makes the tendency more visible.
NO INTERPRETATION OF IRREGULARITY. There is no ready interpretation yet. One interesting feature is the periodical character of this irregularity which rejects earlier explanations based on the random character of initialization. Another possible explanation is the use of Gamma functions in LDA that may produce these periodical fluctuations, however, Gamma functions are not used in pLSA. Therefore, this phenomenon needs more investigation.
MINIMUM IN FIG 6 IS RISKY SELECTION – now it is Fig.8 and it is supplemented with two-dimensional figures 6 and 7 that allow to see the global entropy minimum and the trend more clearly.
4. UNCLEAR VALUES OF q AND r. We define q:= 1/T (line 353), i.e. the inverse number of topics, and r is equal to Jaccard index (line 363).
5. S-M ENTROPY GIVES SMALLER VARIATION THAN RENYI. No, the variation is the same, but for S-M entropy it had been obscured by larger peaks. This disadvantage is easily overcome by re-scaling the graph (see newly introduced fig. 6, 7 and 15), while S-M entropy possesses some unique advantages absent in Renyi entropy. Renyi entropy is a particular case of Sharma-Mittal entropy (when r=1), therefore Sharma-Mittal entropy possesses all properties of Renyi entropy. Moreover, Sharma-Mittal entropy extends Renyi entropy allowing to account for semantic stability of the algorithms of topics modeling that, in turn, is important for users. Renyi entropy does not include comparison of top-words for topic solution with different values of parameters.
6. NO LIMITATIONS OF THE STUDY IN DISCUSSION SECTION. Limitations of our study are added in Discussion (lines 536-552).
7. SIMILAR TITLE. The title was changed.
Additionally, one can find the results of our numerical experiments here: https://yadi.sk/d/4uF8BNxf79q0HQ
Round 2
Reviewer 1 Report
The authors have done an excellent job addressing my concerns and I look forward to seeing this article in print.
Reviewer 2 Report
The authors improved the manuscript and responded my concerns.